# Effect of the Intake of a Traditional Mexican Beverage Fermented with Lactic Acid Bacteria on Academic Stress in Medical Students

**DOI:** 10.3390/nu13051551

**Published:** 2021-05-05

**Authors:** Laura Márquez-Morales, Elie G. El-Kassis, Judith Cavazos-Arroyo, Valeria Rocha-Rocha, Fidel Martínez-Gutiérrez, Beatriz Pérez-Armendáriz

**Affiliations:** 1Biological Science Department, Universidad Popular Autónoma del Estado de Puebla, Puebla 72410, Mexico; laura.marquez.morales@gmail.com (L.M.-M.); eliegirgis.elkassis@upaep.mx (E.G.E.-K.); valeriamagali.rocha@upaep.mx (V.R.-R.); 2Social Science Department, Universidad Popular Autónoma del Estado de Puebla, Puebla 72410, Mexico; judith.cavazos@upaep.mx; 3Center for Research in Health Sciences and Biomedicine, Faculty of Chemical Science, Universidad Autónoma de San Luis Potosí, San Luis Potosi 78290, Mexico; fidel@uaslp.mx

**Keywords:** aguamiel, academic stress, lactic acid bacteria, dysbiosis, gut microbiota, medical students

## Abstract

Dysbiosis of the gut microbiota has been associated with different illnesses and emotional disorders such as stress. Traditional fermented foods that are rich in probiotics suggest modulation of dysbiosis, which protects against stress-induced disorders. The academic stress was evaluated in medical students using the SISCO Inventory of Academic Stress before and after ingestion of an aguamiel-based beverage fermented with *Lactobacillus plantarum*, *Lactobacillus paracasei* and *Lactobacillus brevis* (n = 27) and a control group (n = 18). In addition, microbial phyla in feces were quantified by qPCR. The results showed that the consumption of 100 mL of a beverage fermented with lactic acid bacteria (3 × 10^8^ cfu/mL) for 8 weeks significantly reduced academic stress (*p* = 0.001), while the control group (placebo intervention) had no significant changes in the perception of academic stress (*p* = 0.607). Significant change (*p* = 0.001) was shown in the scores for environmental demands, and physical and psychological factors. Consumption of the fermented beverage significantly increased the phyla Firmicutes and Bacteroidetes but not Gammaproteobacteria. No significant changes were found in the control group, except for a slight increase in the phylum Firmicutes. The intake of this fermented beverage suggest a modulation of gut microbiota and possible reduction in stress-related symptoms in university students, without changing their lifestyle or diet.

## 1. Introduction

Various studies have shown that gut microbiota are linked to stress due to the constant, bidirectional communication on the microbiota–gut–brain axis [1,2,3]. However, the totality of communication pathways has yet to be detected, which is a challenge for future studies. Synbiotics, probiotics and prebiotics appear to have the ability to modulate microbiota, which have a close relationship with axis communication and its effect on stress reduction.

Gut microbiota are seen to be altered by stress and produce communication inconsistencies on the gut microbiota–brain axis when stress becomes chronic [4]. Experiencing stress has been shown to release cortisol, which activates the hypothalamic–pituitary–adrenal axis (HPA) [5]. The hypothalamus sends signals that activate the secretion of corticotropin releasing hormone (CRH), which, in turn, triggers the release of the adrenocorticotropic hormone (ACTH) leading to the secretion of glucocorticoids in the adrenal cortex [6]. As a result, norepinephrine and dopamine are released [7], causing an imbalance in the composition of the gut microbiota and, with that, in the gut bacterial phyla, and neuronal alterations [8,9].

The gut microbiota can be defined as the community of microorganisms and viruses, whose concentration is estimated to be approximately in the order of 10^13^–10^14^ and which interact synbiotically in the human intestine [10,11,12]. Representative phyla of the gut microbiota include Firmicutes, Bacteroidetes, Proteobacteria, Actinobacteria, Archaea, Fusobacteria and Verrucomicrobiota [13,14,15]. Previous studies have shown that the genera *Lactobacillus* and *Bifidobacteria*, belonging to the phyla Firmicutes and Actinobacteria, respectively, are producers of gamma-aminobutiric acid and serotonin [16], both of which are altered in stressed individuals [17], causing dysbiosis characterized by the growth of enterobacteriaceae capable of producing norepinephrine [18,19]. This leads to a stress feedback loop. These bacterial changes in the gut microbiota can lead to depression and anxiety [20,21], impairment of memory and reasoning [22,23], sleep disorders [24] and irritable bowel syndrome [25], all of which have been detected in university students.

In recent years, medical students have been reported to present particularly high stress levels due to being exposed to many academic demands [26,27,28]. Preclinical studies have shown that stool samples, obtained from university students after a high-stress situation such as an examination period, evidenced a marked reduction in lactic acid microorganisms [29,30] (Figure 1). Another clinical study showed that subjects suffering from stress-related irritable bowel syndrome presented an increase in the phyla Proteobacteria and Barnesiella, but a decrease in Firmicutes and Bacteroidetes [31]. In addition, people in a state of depression—a mental illness that, on many occasions, presents in stressed students—showed an increase in Bacteroidetes, Proteobacteria and Actinobacteria and a decrease in Firmicutes [32]. In animal studies, dysbiosis has also been observed in gut microbiota when subjects were exposed to social disruption [33,34].

Recent studies have sought to restore balance to the gut microbiota through synbiotic foods that appear to have a beneficial effect on human health and the neuronal functions of the brain [35]. Synbiotics are a mix of probiotics and prebiotics. Probiotics are living microorganisms which, at the right dose, improve health. Prebiotics, in contrast, are substances which cannot be digested by the body, but which induce the growth and activity of probiotics [36].

Earlier studies of interventions with *Lactobacillus casei Shirota* probiotics have demonstrated a reduction in stress as well as favorable effects on intestinal dysfunction problems and changes in the microbial diversity of the gut [37]. Analysis of the consumption of the synbiotic Ecologic^®^825, which contains different strains of *Lactobacillus* and *Bifidobacteria,* evidenced changes in emotional memory, reasoning and emotional decision-making [38], abilities that are diminished under stress. Meanwhile, sleep quality and symptoms of anxiety and depression in stressed students were improved with the use of heat-activated *Lactobacillus gasseri* strain CP2305. Furthermore, an increase was shown in the genus *Bifidobacterium*, as well as a reduction in *streptococci* [39]. On the other hand, nine lyophilized strains of the genera *Lactobacillus* and *Bifidobacterium*, in the synbiotic Ecologic^®^Barrier, improve cognition in depressed university students, without reducing depression [40]. Recently, there has been a great interest in demonstrating the relationship of prebiotic and probiotic consumption on mental health, as shown in Table 1.

Previous studies have shown that traditional fermented foods are a good source of probiotics and prebiotics that can modulate the gut microbiota [48,49]. In Mexico, pulque, a pre-Hispanic millennial drink, is a good source of lactic acid bacteria (LAB) [50,51]. Aguamiel is rich in fiber, which acts as a prebiotic, and contains fructooligosaccharides—short polymers of fructose (inulin)—which promote the survival of probiotics in the colon [52]. Aguamiel can also help regulate the gut microbiota [53]. There are few studies of experimental interventions with synbiotics made from products such as pulque and Aguamiel, particularly in stressed university students.

The aim of this research was to assess the effect of a traditional beverage (Aguamiel) fermented with lactic acid bacteria (*L. plantarum, L. paracasei* and *L. brevis*) on stressed medical students using the SISCO Academic Stress Questionnaire and a qPCR assay of three phyla in the microbiota of feces to produce healthy regional food alternatives and mitigate the effects of stress. It is important to note that *Lactobacillus plantarum, Lactobacillus paracasei* and *Lactobacillus brevis* were recently reclassified as *Lactiplantibacillus plantarum, Lacticaseibacillus paracasei, Levilactobacillus brevis*, respectively [54].

## 2. Materials and Methods

### 2.1. Sampling and Study Design

This clinical trial was approved by the Research Ethics Committee of the Health Sciences department of the Popular Autonomous University of the State of Puebla (CONBIOETICA21CEI00620131021), Puebla, Mexico, and followed the 1975 Helsinki protocol on human experimentation standards. A single blind, randomized, longitudinal, prospective, experimental, controlled study was conducted. The methodology was divided into two phases: Phase I, elaboration of the fermented beverage with lactic acid bacteria (FBLAB) and elaboration of the placebo, and Phase II, the intervention. The sample comprised 52 male and female students between the ages of 20 and 25. All were Mexican and studying medicine at the private university in the city of Puebla, Mexico. The sample was split into two groups: the experimental group (EXP), *n* = 27, who consumed the FBLAB, and the control group (CTL), *n* = 18, who consumed the placebo beverage. Levels N1 and N2 were considered low stress; levels N3, N4 and N5 high stress. Students with a history of anxiety, bipolar disorder, epilepsy, schizophrenia, insomnia or any other neurological disorder, and pregnant women, were excluded. In addition, anyone who had consumed any kind of drug in the four months prior to the trial and/or was under treatment with antidepressants was excluded. The product was delivered once a week, the bottles were labeled with the name of the day (Monday, Tuesday, etc.) and it was ensured that the participants followed the directions of the exclusion criteria. Furthermore, other aspects of the lifestyle were not controlled (such as smoking, eating, drinking and exercising) to evidence the effects of consumption of the FBLAB under conditions similar to real life. Students who did not provide the information necessary for the measuring instruments or who did not finish the treatment were withdrawn from the study. Participation was voluntary. An informed consent form, explaining the study in detail, was signed. Study participants received no compensation or economic support.

### 2.2. Phase I. Elaboration of the FBLAB and Placebo Beverages

#### 2.2.1. Isolation and Selection of Microorganisms from Pulque

Three microorganisms of the genus *Lactobacillus* were isolated from samples of pulque acquired in San Pedro Cholula, Puebla, Mexico. A seed culture was made with 0.1 mL of pulque in 10 mL of Mann Rogosa Sharpe (MRS) broth medium (Merk, Kenilworth, NJ, USA) and incubated at 37 °C under anaerobic conditions for 48 h. The decimal dilution technique and cross-streaking on MRS agar medium was used until pure strains were obtained [55]. The purified strains were then evaluated with the Gram stain and catalase test to prove that they were lactic acid bacteria. In addition, microbial growth was quantified through growth kinetic studies for 48 h (turbidimetry). The analysis was conducted using UV visible spectrophotometry (model 4255, Seville, Spain) at 560 nm.

#### 2.2.2. Identification of Isolated Strains with LAB Characteristics

The strains used in this study were identified by MALDI-TOF (Matrix Assisted Laser Desorption Ionization-Time of Flight) mass spectrometry, with 2.2 software (Bruker Daltonics, Billerica, MA) (Bruker, Germany). The isolated strains had the following scores based on mass spectrometry identification: 2.078 *Lactobacillus plantarum*, 2.374 *Lactobacillus paracasei* and 2.23 *Lactobacillus brevis.* Bacterial DNA extraction was performed in triplicate from a 5 mL culture of each of the three strains, using the Quick DNA Fecal/Soil Microbe Miniprep kit (Zymo Research, Irvine, CA, USA), following the manufacturer’s instructions. The integrity and quantity of extracted DNA was verified using the Genova NanoDrop (Jenway™, Bibby Scientific, Burlington, NJ, USA) spectrophotometer to quantify the DNA, measuring absorbance at 260 nm and 280 nm.

#### 2.2.3. Elaboration of FBLAB

The Aguamiel was pasteurized at 80 °C for 30 min and sucrose added to bring its content to 10° Brix. When the beverage was at room temperature, a mixed bacterial inoculum made up of *L. plantarum, L. paracasei and L. brevis* was added at a concentration of 1%, with an ODλ_600 nm_ between 0.6 and 0.8, and fermented for 14 h. The beverage must reach a probiotic concentration of 3 × 10^8^ cfu/mL. Subsequently, a previously pasteurized coconut flavoring (COCO DREAM^®^) was added. Packaging was conducted under sterile conditions using previously pasteurized containers and conserved at 4 °C. The detailed methodology was in accordance with the provisions of patent No. 371480 (Mexico) [56].

#### 2.2.4. Elaboration of the Placebo Beverage

The placebo beverage was made from coconut water and flavoring (COCO DREAM^®^). The process consisted of pasteurizing the coconut water at 80 °C for 30 min, before the Brix degrees were standardized at 10 and flavoring was added at a concentration of 0.2%. The beverage was packaged and conserved under the same sterile conditions as the probiotic beverage.

### 2.3. Phase II. Student Intervention with FBLAB and the Placebo Beverage

#### 2.3.1. Trial Design

The trial was experimental, longitudinal, single-blind, exploratory, with non-randomized sampling. The beverages were delivered to the students for the consumption of 100 mL per day for a period of two months from 10 April to 10 June 2019. The administration of the beverages was monitored through reminder messages and verification of consumption. The SISCO questionnaire was applied before and after the intervention. Stool samples were collected from each study subject before and after the intervention to determine the gut microbiota.

#### 2.3.2. Instrument

The instrument used in this study was the SISCO Inventory of Academic Stress. This survey is an instrument proposed and validated by Barraza [57] for the study of chronic academic stress. It measures the adverse effect of stress on the behavior and health of students and was previously applied in other Latin American countries [58,59].

Data collection was performed by applying the questionnaire in the classroom during the summer academic term. It was applied individually and had a duration of 15 min. The Cronbach’s alpha obtained was 0.90. The instrument consisted of 1 dichotomous filter item that decides whether the respondent will continue with the survey. Stress in general was determined by one question with a five-value Likert-type scale (1 to 5, where 1 is low and 5 is high) to identify the intensity of academic stress. This was followed by 29 items, which are answered using a 5-value Likert-type scale (Never, Almost Never, Sometimes, Almost Always, and Always). These items are divided into three sections: Stressors (8 items), Symptoms (15 items) and Coping Strategies (6 items) [57,60].

#### 2.3.3. Statistical Method

Measures of central tendency and dispersion were calculated for numerical variables; counts and percentages were calculated for categorical variables. Independent Student’s t and $\chi^{2}$ tests were performed. Test results were processed using the SPSS program, version 23 (SPSS Inc., Chicago, IL, USA) with *p* < 0.05 considered significant.

### 2.4. Analysis of Gut Microbiota

#### 2.4.1. Collection of Stool Samples

Stool samples were collected from all subjects on two occasions: the first, one week after applying the questionnaire and before the intervention. The second stool sample was collected after the intervention in the two groups (EXP and CTL). The samples were collected in sterile H1015S sample cups (Vela Quin S de R.L de C.V) and immediately frozen at −70 °C, where they remained until analysis.

#### 2.4.2. Bacterial DNA Extraction

The microbial DNA extraction from the stool samples was performed in triplicate using the QuickDNA Fecal/Soil MiniPrep kit (Zymo Research, Irvine, CA, USA) according to the manufacturer’s instructions. Quantification of nucleic acids was conducted by means of UV-visible spectrophotometry, measuring absorbance at 260 nm and 280 nm in a spectrophotometer (Genova NanoDrop Jenway™, Bibby Scientific, Burlington, NJ, USA).

#### 2.4.3. Quantification of Phyla Bacteroidetes, Firmicutes and Gammaproteobacteria by qPCR

Quantification of the phyla Firmicutes, Bacteroidetes and Gammaproteobacteria was conducted by amplification of *16S rRNA* gene by real-time PCR in a Rotor-Gene Q thermocycler using the Rotor-Gene SYBR Green PCR kit (Qiagen, Germantown, MD, USA). The primer used are: Firmicutes (Firm934F) GGAGYATGTGGTTTAATTCGAAGCA, (Firm1060R) AGCTGACGACAACCATGCAC [61]; Bacteroidetes (Bac960F) GTTTAATTCGATGATACGCGAG (Bac1100R) TTAASCCGACACCTCACGG [62]; Gammaproteobacteria (γ1080F) TCGTCAGCTCGTGTYGTGA, (γ1202R) CGTAAGGGCCATGATG [63].

The PCR reactions were performed in triplicate in a total volume of 25 µL to which 1 ng of bacterial genomic DNA was added. The reaction volumes were: 7.5 µL of SYBR Green Master Mix 2X, 1 µL forward primer (10 μM) and 1 ul reverse primer, 10 uM and 4.5 µL RNAase-free water and 1 µL genomic DNA (1 ng/μL). The SybrGreen^®^ PCR kit (Qiagen, Germantown, MD, USA) was used.

The PCR program used in the case of the phylum Bacteroidetes is as follows: (1) 95 °C for 5 min, (2) 95 °C for 15 s, (3) 64 °C for 15 s, (4) 72 °C for 4 s, (5) repeat stages 2, 3 and 4, 45 times, (6) 4 °C for 10 min. The PCR program used for the phylum Firmicutes is the following: (1) 95 °C for 5 min, (2) 95 °C for 15 s, (3) 61 °C for 15 s, (4) 72 °C for 10 s, (5) repeat stages 2, 3 and 4, 45 times, (6) 4 °C 10 min. The PCR program used for the phylum Gammaproteobacteria is the following: (1) 95 °C for 5 min, (2) 95 °C for 15 s, (3) 60 °C for 15 s, (4) 72 °C for 10 s, (5) repeat stages 2, 3 and 4, 45 times, (6) 4 °C for 10 min.

The calibration curve for the calculation of the number of copies of *16S RNAr* gene of the phyla of interest was prepared by cloning one *16S ARNr* gene from a representative of each phylum of interest into known quantities of the plasmid pMiniT 2.0 (New England Biolabs, Ipswich, MA, USA). These genes were amplified using specific primers: Bacto0297 and Bacto1245 for Bacteroidetes [64], BacF and R1378 for Firmicutes [65] and Bacecofw 1428 and Bacecorev for Gammaproteobacteria [66].

#### 2.4.4. Percentage Change in Abundance of the Phyla

Determination of the change in abundance of the phyla Bacteroidetes, Firmicutes and Gammaproteobacteria after treatment compared to before treatment was calculated using the following formula:

% change in abundance = [(TA−TB)/TA] * 100

TA = Total number of copies after consumption of FBLAB or placebo

TB = Total number of copies before consumption of FBLAB or placebo

#### 2.4.5. Statistical Analysis of Changes in Abundance of the Phyla

The means of the number of copies in each phylum were calculated and minimum and maximum values determined at each measurement time and in each intervention group. Non-parametric Kruskal–Wallis tests were applied to make comparisons between the three phyla; the Wilcoxon signed rank test was used to compare the before and after measurements and the Mann–Whitney test was used to compare the measurements between the intervention groups. The analysis was performed with the GraphPad Prism program, version 8.0.0, for Windows (GraphPad Software, San Diego, CA, USA) Statistical significance was established at *p* ≤ 0.05.

Correlation tests were performed, using Pearson’s test, to measure correlations between stress levels and the abundance of the *16s rRNA* genes of the studied phyla. Significance was determined at *p* ≤ 0.05.

## 3. Results

### 3.1. Academic Stress Level of Students before and after Intervention with FBLAB and Placebo

Of the sample of 52 students, 7 were dropped based on the elimination criteria. The information of the remaining 45 participants was analyzed, of whom 60% were women and 40% men between the ages of 20 and 25 years. A descriptive analysis of the level of concern or nervousness was carried out in all the medical students. All the medical students included in the sample reported academic stress. The students were divided into two groups, the CTL group (*n* = 18) with an average level of stress of 2.88, and the EXP group (*n* = 27) with an average stress level of 3.88. This difference is a limitation for the comparison between groups; therefore, it must be considered that at the beginning of the intervention, the EXP group expressed significantly higher levels of stress (*p* = 0.001). However, it is possible to identify the change (before and after) for each group independently.

Figure 2a presents the results of the measurement of academic stress for the CTL group; stress perception showed no significant difference (*p* = 0.607) before and after interventions, while, for the EXP group, the stress level showed a significant reduction (*p* = 0.001) (Figure 2b) after interventions. The results show that, in the EXP group, 0% of participants reported high stress levels (N4 and N5) after the intervention, compared to 22.2% of participants in the CTL group, while 77.8% of participants in the EXP group reported low stress levels (N1 and N2) after the intervention, compared to 22.2% of participants in the CTL group.

Regarding to the results for environmental stressors using the SISCO questionnaire for paired samples (Student’s T-test for paired samples), the results for the EXP group indicated a significant change in the scores of all the variables, except for the religiosity variable (*p* = 0.422) (Table 2).

In the CTL group, 25 of the 29 stressors were non-significant while four were significant, presenting lower means in the second measurement: two environmental stressors (competition with fellow students (*p* = 0.018) and excessive assignments and schoolwork (*p* = 0.013)), one physical reaction (scratching, nail-biting, rubbing, etc. (*p* = 0.014)), and one psychological reaction (anxiety, anguish, despair (*p* = 0.001)).

In the Student’s *T*-test for independent samples (Table 2), the mean stress scores before and after the interventions were compared. In the EXP group, the mean stress score of 12 items showed a significant decrease.

### 3.2. Effect of FBLAB Consumption on Gut Microbiota

The effect of the consumption of the LAB-fermented beverage was evaluated on three phyla, Bacteroidetes, Firmicutes and Gammaproteobacteria, in the gut microbiota of stressed students. The results presented in Figure 3 show the number of copies of the *16S rRNA* gene of the studied phyla. The ANOVA analysis of the phylum Bacteroidetes showed a significant increase in the mean number of copies of the *16S* gene in the stools of the EXP group after intervention (*p* = 0.0001), while in the CTL group, no significant differences were found in the mean number of copies of the same gene before and after intervention (*p* = 0.5798). The phylum Firmicutes showed a significant increase in the mean number of copies of the *16S* gene in stools of the EXP group after intervention (*p* = 0.0056); in addition, the CTL group showed a significant increase in the firmicutes (*p* = 0.0385). The increase in the phylo firmicutes in both studied groups, EXP and CTL, showed different meanings. For the EXP group, the dysbiosis in the ratio of firmicutes to bacteroidetes decreased because both bacteroidetes and firmicutes increased, while, for the CTL group, the dysbiosis was more marked, because the abundance of the phylo bacteroidetes did not increase in the same proportion (Figure 3a,b). The phylum Gammaproteobacteria showed no significant differences (Figure 3c) in the mean number of copies of the *16S* gene before and after intervention in both the EXP (*p* = 0.0731) and CTL (*p* = 0.2121) groups.

Due to the great natural variability between individuals in the number of bacteria of each phylum in the gut microbiota, we chose to present the results as a percentage of change in each subject before and after consumption of the FBLAB and the placebo. The results are shown in Figure 4. A significant increase in the abundance of Bacteroidetes (83%) was observed after consumption of the FBLAB in the EXP group, but no significant change in the abundance of this phylum (3%) was observed after consumption of the placebo. The consumption of both beverages (FBLAB and placebo) resulted in an increase in the abundance of the phylum Firmicutes for both groups; however, consumption of the FBLAB generated a significantly higher increase (95%) in the abundance of Firmicutes than consumption of the placebo (19%). No significant change was observed in the abundance of the phylum Gammaproteobacteria in gut microbiota after consumption of either beverage (6% for the EXP group and 4% for the CTL group).

In Table 3, the correlation between stress levels and observed changes in studied microbial phyla was determined using Pearson test. A statistically significant negative correlation was observed between stress levels and the abundance of firmicutes in gut microbiota of the EXP group before the consumption of the FBLAB.

## 4. Discussion

The results of the Figure 1 are consistent with a study where a probiotic made with *L*. *plantarum* DR7 administered to stressed adults for 12 weeks reduced stress symptoms along with a significant decrease in plasma pro-inflammatory cytokines such as IFN-γ (*p* < 0.0001) and TNF-α (*p* = 0.0006) in the DR7 group as compared to the placebo group [67]. Regarding to the results for environmental stressors using the SISCO questionnaire for paired samples (Student’s T-test for paired samples), the results for the EXP group indicated a significant change in the scores of all the variables, except for the religiosity variable (*p* = 0.422) (Table 2). This coincides with other studies where it has been observed that medical students value these items more frequently as stressors [68,69,70]. The results indicated a significant association between the consumption of FBLAB and environmental demands and physical, psychological and behavioral reactions, as well as six coping responses, showing lower means after intervention. Regarding physical symptoms, consumption of FBLAB generated positive effects similar to the probiotic *L*. *casei Shirota* (Lcs), reported as having a beneficial effect on sleep quality after states of high stress. This study showed a 20% reduction in delta power (initial time to sleep and final time), in the probiotic group versus the placebo group, after 11 weeks of the intervention, concluding that the consumption of Lcs may help to maintain the perceived quality of sleep during the stress period [48]. Other studies with multispecies probiotics showed a reduction in symptoms, such as diarrhea and irritable bowel syndrome (IBS), that are associated with stress, showing a significant decrease (*p* < 0.0001) with the consumption of *B. bifidum* R0071, as compared to placebo and other probiotics. In this study, the EXP group showed a significant reduction (*p* = 0.001) in the occurrence of diarrhea and IBS, unlike the CTL group, which showed no change (*p* = 0.302) after the intervention. The EXP group showed improvement in the item “chronic fatigue”, while the CTL group showed no change (Table 2). Furthermore, the interventions with FBLAB in this study show that it has a favorable effect on psychological reactions including depression (*p* = 0.001). Similarly, other research showed that using *L. plantarum* 299v (LP299) reduces symptoms of depression. Rudzki et al. [71], showed a significant reduction in the expression of kynurenine (a metabolite associated with depression, involved in the tryptophan pathway) in people who consumed LP299.

In terms of behavioral reactions, the EXP group showed changes in eating habits, with the item “increase or decrease in food” being significant. The consumption of a probiotic with *L*. *rhamnosus* has been shown to reduce stress symptoms, appetite and cravings and improve self-esteem in women [72,73]. Stress has been shown to reduce the ability to cope with problems; however, it has been proven that the administration of single or multispecies probiotics can reduce stress and help improve cognitive ability to cope with situations common to students [74]. This suggests that the consumption of FBLAB in this study shows similar effects, as observed in most of the items. However, more studies are required using other support instruments.

The increased abundance of Firmicutes in the gut microbiota of the EXP group after invention with FBLAB may be a consequence of the daily consumption of a beverage fermented with three species of the genus *Lactobacillus* belonging to the phylum Firmicutes (Figure 3). Previous studies have related the consumption of strains of the genus *Lactobacillus* with increased abundance of Firmicutes in gut microbiota [75,76]. The increase observed in the genus Bacteroidetes after consumption of the FBLAB is in contrast with a previous study, which showed that consumption of the strain *L. Casei Shirota* had no significant effect on this phylum in the gut microbiota of people with obesity [77]; these differences may be due to the fact that, in this study, subjects suffered from stress and not obesity. The balance of the microbial *phyla* is multifactorial: gender, stress, obesity, doses and intervention time are some of the factors that modulate the gut microbiota. In addition, several mechanisms are involved in the balance of the gut microbiota such as the increase in immune cells, the permeability of the intestinal barrier and the presence of antimicrobial peptides [76,77]. The observed increase in the abundance of both Firmicutes and Bacteroidetes (Figure 4) makes it possible to maintain a balance between these two phyla, a balance usually associated with a healthy gut microbiota [78] and possibly with the observed decrease in stress (Figure 2). The increase in Bacteroidetes shows an equilibrium with Firmicutes (Figure 4), which may suggest that the Firmicutes Bacteroidetes balance is associated with decreased stress. The absence of effects in the EXP group on the abundance of Gammaproteobacteria coincides with a previous study where no significant change was observed in the abundance of *Enterobacteriaceae* after consumption of *L. acidophilus* and *Bifidobacterium animalis*. However, although Gammaproteobacteria did not show a change, it is recognized that small changes in the composition of the fecal microbiota affect gene expression and metabolic products [78].

These results suggest that the consumption of the FBLAB restored balance to the gut microbiota of stressed students, as shown by the increased abundance of Bacteroidetes and Firmicutes (Figure 4). This change is likely to be the result of an increased abundance of beneficial species belonging to both groups of microorganisms. It is important to highlight that while the abundance of Gammaproteobacteria did not change significantly after consumption of the FBLAB (EXP group), its proportion of total bacteria decreased in relation to the significant increase in Bacteroidetes and Firmicutes. The decrease in stress levels of medical students after consumption of the FBLAB suggests an association with the restoration of a healthy gut microbiota.

Different results have been obtained for intervention with probiotics, prebiotics or synbiotics and stress in a study by Gautam [79], which observed an increase in the ratio of firmicutes/bacteroidetes with increased stress levels in rodents, as opposed to results when interventions are made in humans [80]. In general, an increase in the proportion of firmicutes and a decrease in the proportion of bacteroidetes in the gut microbiota are associated with several diseases such as obesity [81]. However, an increasing body of knowledge indicates that this is not always the case since diverging results are reported under varying circumstances such as disease type, age, geographical origin of individuals, lifestyle, diet, etc. [80]. When stress levels in the EXP group before and after the treatment were compared, a statistically significant negative correlation was observed between stress levels and the abundance of both firmicutes and bacteroidetes in gut microbiota (Table 3). The reduced microbial diversity in the gut increases the risk of a negative health impact from the pathogenic bacteria. The dysbiosis generates an increase in the permeability of the intestinal barrier and the reduction in the immune response, resulting in a bacterial translocation and a possible negative impact on the brain–gut axis [11]. Our results highlight the need for a more detailed study of the gut microbiota at the family and genus levels. This would greatly improve our understanding of the complex dynamics of gut microbiota in response to diet, lifestyle and environmental stimuli and its relationship with human health.

## 5. Conclusions

The consumption of the FBLAB—a mead-based beverage fermented with *L. plantarum, L. paracasei and L. brevis*—may have a beneficial effect on stress reduction and the correction of the dysbiosis in the gut microbiota of university students, without interfering with their lifestyle or diet.

Traditional or new lactic acid beverages can be a healthy food option for gut microbiota modulation. It is necessary to repeat this study with a larger group of participants, and this will be conducted in the future to confirm the promising results identified in this study regarding the effectiveness of the FBLAB.

## Figures and Tables

**Figure 1 nutrients-13-01551-f001:**
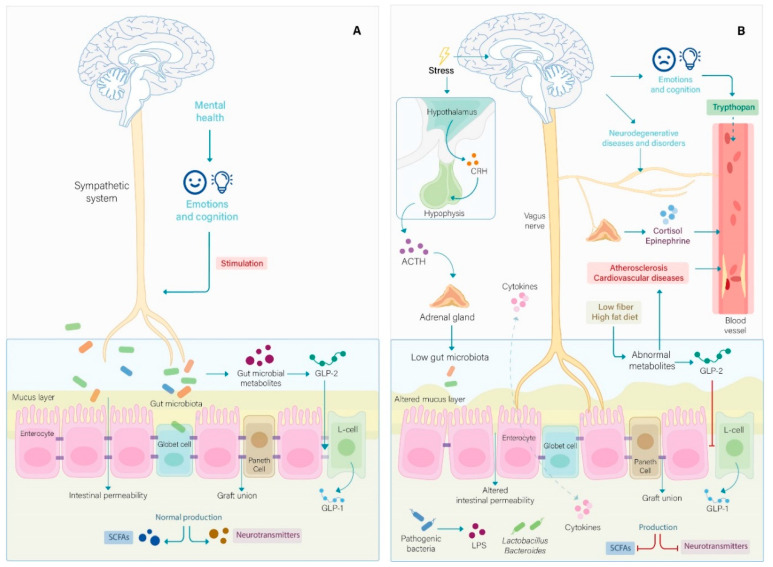
(**A**) The healthy microbiota mainly indicates genera *Lactobacilli* and *Bacteroides* that maintain a normal production of short chain fatty acids (SCFAs) and neurotransmitters such as serotonin, giving a better response to stress and returning to the basal state. The intestinal microbiota, when in balance (Eubiosis), allow an interaction with the vague nerve that maintains the normal metabolism of tryptophan, a precursor of serotonin, and the secretion of gamma aminobutyric acid to generate a healthy psychological and emotional state, in addition to maintaining adequate intestinal secretion and motility. The reduction in pathogenic bacteria in the intestine is controlled by the beneficial microbiota, and the secretion of antimicrobials by the Paneth cells increases the production of glucagon-like peptide (GLP1) that exerts beneficial effects on the metabolism of glucose and glucagon-like peptide-2 (GLP2) that maintains the integrity of the membrane. (**B**) Dysbiosis refers to a reduced community of beneficial bacteria and an increased number of pathogenic bacteria. When there is stress, the HPA axis and the sympathetic system are deregulated; therefore, the adrenal glands maintain elevated levels of cortisol, epinephrine, and norepinephrine; this leads to a constant feedback to stress with the reduction in SCFAs and the alteration of neurotransmitter levels. The heart rate and energy requirements increase, causing the consumption of a caloric diet. The low-fiber, high-fat diet causes the fermentation of metabolites by the intestinal microbiota to be abnormal. The production of the GLP2 peptide is decreased; therefore, intestinal permeability is increased; this allows the increase in pathogenic bacteria and lipopolysaccharides (LPS), causing loss of intestinal function.

**Figure 2 nutrients-13-01551-f002:**
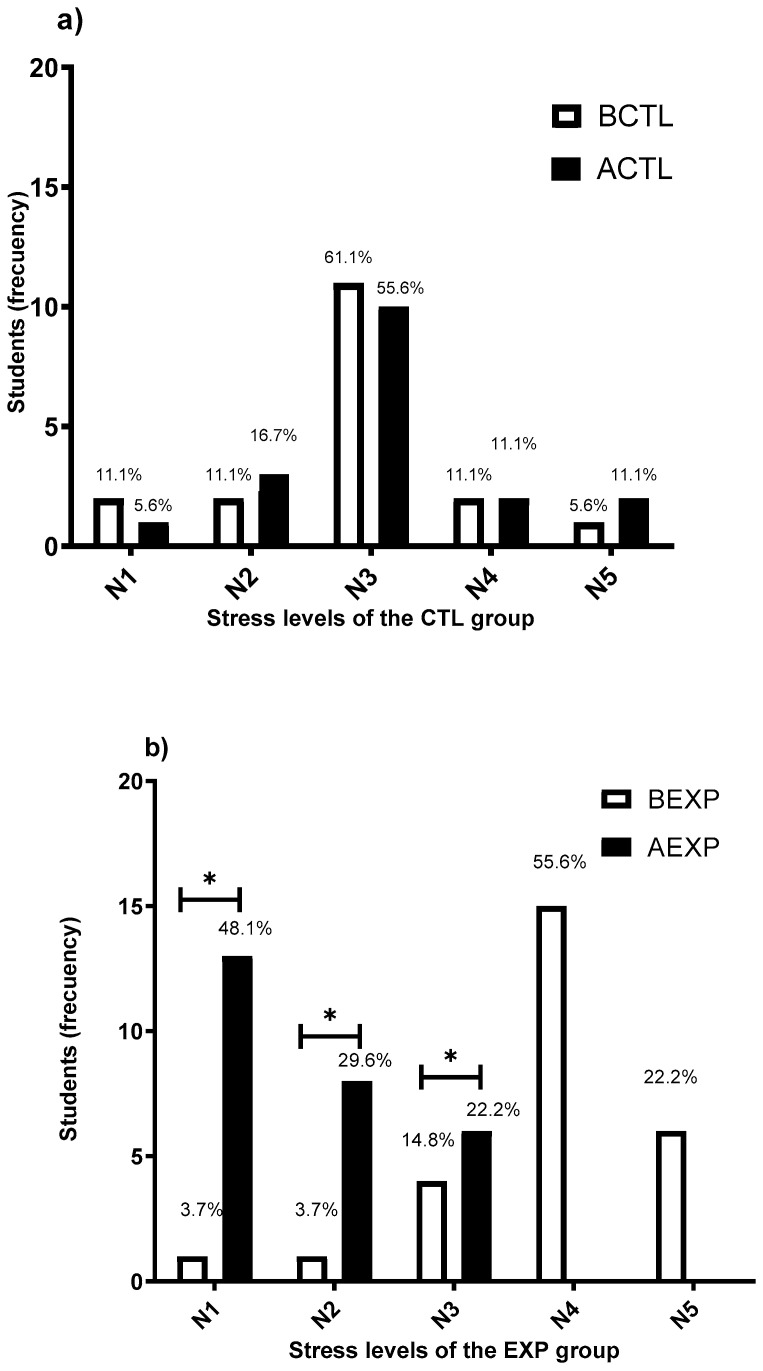
Academic stress intensity levels. Frequency of students which perceived each stress level: N1, N2 (low stress level), N3, N4, N5 (high stress level). (**a**) Before (BCTL) and after (ACTL) the intervention with the placebo beverage (CTL group): data did no show significant differences (*p* = 0.607). (**b**) Before (BEXP) and after (AEXP) the intervention with the FBLAB (EXP group): data showed significant differences (* *p* = 0.001, α = 0.05). Statistical analysis was performed using the Student *t*-test.

**Figure 3 nutrients-13-01551-f003:**
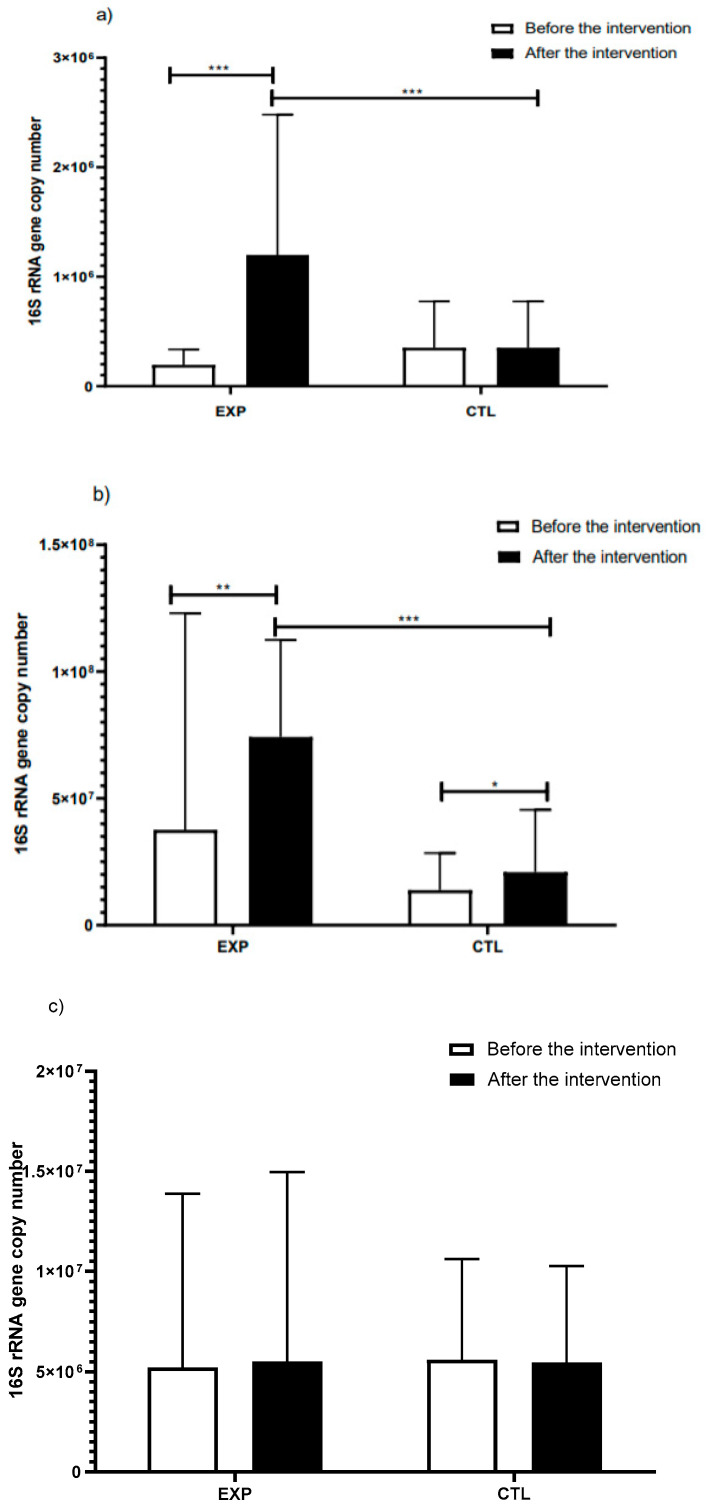
Abundance of the phyla Bacteroidetes (**a**), Firmucutes (**b**) and Gammaproteobacteria (**c**), in the feces microbiota of students with academic stress before (white) and after (black) the intervention with the FBLAB (EXP group, *n* = 27) or the placebo beverage (CTL group, *n* = 18). Abundance is expressed as *16S rRNA* gene copy number. Non-parametric Kruskal–Wallis tests were applied to make comparisons between the three phyla; the Wilcoxon signed rank test was used to compare the before and after measurements and the Mann–Whitney test was used to compare the measurements between the intervention groups. Significant difference: * *p* < 0.05, ** *p* < 0.01, *** *p* < 0.001.

**Figure 4 nutrients-13-01551-f004:**
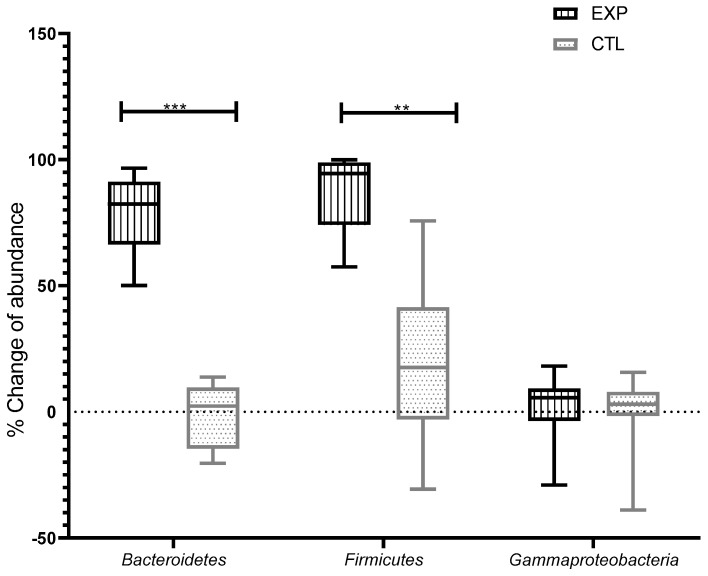
Box-and-whisker plot representing percentages of change of abundance of the phyla bacteroidetes, firmicutes and gammaproteobacteria in feces microbiota of students with academic stress after the intervention with the BFLAB (EXP group, vertical lines) or the placebo beverage (CTL group, dots), compared to before the intervention. The percentage of change in abundance was calculated using the following formula: % change in abundance = [(TA−TB)/TA] × 100. TA = number of *16S rRNA* copies after consumption of FBLAB or placebo. TB = number of *16S rRNA* copies before consumption of FBLAB or placebo. Statistical analysis was performed using the Student’s t-test. ** *p* < 0.01, *** *p* < 0.001.

**Table 1 nutrients-13-01551-t001:** Interventions with probiotics, prebiotics and synbiotics in stressed individuals.

Disease or Disorder (Country)	Product	Treated Subjects	Probiotics	Prebiotics	Beneficial Health Effects	Time	References
Stress (Japan)	Probiotic	23 men and 25 women (˂30 years)	*L. casei shirota* YIT 9029 1 × 10^11^ CFU/mL	Fermented milk	Decreased abdominal dysfunction, abdominal pain, feelings of stress, cortisol reduction.	8 weeks	[37]
Anxiety, depression, stress (Japan)	Synbiotic	29 medical students, both sexes	*Lactobacillus gasseri* CP2305 1 × 10^10^ cells	Fermented milk CP2305	Improves symptoms associated with stress. Increase in *Bifidobacteria* and decrease in *Streptococci*	24 weeks	[39]
Insomnia (Japan)	Para-psychobiotic CP2305	33 medical students, both sexes (18–34 years)	*Lactobacillus gasseri* CP2305 1 × 10^10^ cells	Fermented milk CP2305	Favorable effect on physical symptoms and sleep quality associated with stress	5 weeks	[41]
Stress (Japan)	Probiotic	49 4th grade medical students	*Lactobacillus casei shirota* 1 × 10^9^ CFU/mL	Fermented milk	Improves the quality of sleep according to the OSA analysis.	11 weeks	[42]
Stress (Japan)	Probiotic	70 medical students	*Lactobacillus casei shirota* 1 × 10^9^ CFU/mL	Fermented milk	Cortisol reduction, decreased abdominal discomfort, and stress flu. It can control HPA in rats.	8 weeks	[43]
Stress (France)	Probiotic	219 healthy volunteers (18 to 70 years)	*Lactobacillus gasseri* PA*, Bifidobacterium bifidum* MF*, Bifidobacterium longum* SP		Reduction in stress and fatigue	32 days	[44]
Stress (USA)	Probiotic	Undergraduate students: *n* = 145 for *L. helveticus* R0052, *n* = 142 for *B. bifidum* R007, *n* = 147 for *B. infantis* R0033	*Bifidobacterium longum ssp. Infantis* R0033*, Bifidobacterium bifidum R0071 and Lactobacillus helveticus* R0052 3 × 10^9^ CFU		Improvements in symptoms of diarrhea, constipation and abdominal pain.	6 weeks	[45]
Stress (Iran)	Symbiotic	30 women (18–40 years)	*Lactobacillus acidophilus, Lactobacillus casei and Bifidobacterium bifidum* 2 × 10^9^ CFU/g	Inulin	Significant increase in serum sex hormone binding globulin. Reduction in serum insulin levels. Nitric oxide increase. Beneficial effect on C-reactive protein	12 weeks	[46]
Obesity (Iran)	Symbiotic	30 men and 20 women	*Lactobacillus acidophilus, Lactobacillus casei, Bifidobacterium bifidum* 2 × 10^9^ CFU	Inulin	Decrease in body weight, stress, anxiety	8 weeks	[47]

**Table 2 nutrients-13-01551-t002:** Effect of the intervention with BFLAB or placebo beverage on the severity of academic stress-related symptoms, as determined using the SISCO questionnaire ^(1)^.

VARIABLES	CTL Group ^(2)^			EXP Group ^(2)^				Comparison between Groups ^(3)^		
Stressors from Environmental Demands	Before ($\bar{x}$)	After ($\bar{x}$)	*p*	Before ($\bar{x}$)	After ($\bar{x}$)	*p*	Before ($\bar{x}$)	*p*	After ($\bar{x}$)	*p*
Competition with group mates	2.44 ± 0.86	1.67 ± 0.91	0.018 **	2.89 ± 0.80	1.00 ± 1.11	0.001 **	CTL 2.44 ± 0.20 EXP 2.89 ± 0.15	0.089	CTL 1.67 ± 0.21 EXP 1.00 ± 0.21	0.033 *
Home and schoolwork overload	3.67 ± 0.84	2.78 ± 1.35	0.013 **	4.00 ± 0.62	1.41 ± 1.60	0.001 **	CTL 3.67 ± 0.19 EXP 4.00 ± 0.11	0.160	CTL 2.78 ± 0.31 EXP 1.41 ± 0.30	0.004 **
The teacher’s personality and character	2.28 ± 0.83	2.83 ± 1.34	0.078	2.89 ± 0.93	1.70 ± 1.79	0.003 **	CTL 2.28 ± 0.19 EXP 2.89 ± 0.18	0.026 *	CTL 2.83 ± 0.31 EXP 1.79 ± 0.34	0.02 *
Evaluations (exams, essays, research papers, etc.)	3.56 ± 1.10	2.89 ± 1.64	0.175	4.11 ± 0.70	1.30 ± 1.54	0.001 **	CTL 3.56 ± 0.25 EXP 4.11 ± 0.13	0.068	CTL 2.89 ± 0.38 EXP 1.30 ± 0.29	0.002 **
The type of work required by the teachers (consultation of topics, worksheets, essays, concept maps, etc.)	3.06 ± 0.87	2.50 ± 1.29	0.172	3.37 ± 0.84	1.56 ± 1.80	0.001 **	CTL 3.06 ± 0.20 EXP 3.37 ± 0.16	0.237	CTL 2.50 ± 0.30 EXP 1.56 ± 0.34	0.047 *
Not understanding the topics discussed in class.	2.89 ± 1.08	2.50 ± 1.10	0.321	3.44 ± 1.01	1.44 ± 1.60	0.001 **	CTL 2.89 ± 0.25 EXP 3.44 ± 0.19	0.092	CTL 2.50 ± 0.25 EXP 1.44 ± 0.30	0.012 *
Class participation (answering questions, presentations, etc.)	2.39 ± 1.33	2.17 ± 1.42	0.655	3.11 ± 1.05	1.30 ± 1.46	0.001 **	CTL 2.39 ± 0.31 EXP 3.11 ± 0.20	0.063	CTL 2.17 ± 0.33 EXP 1.30 ± 0.28	0.054
Limited time to complete tasks	3.00 ± 1.08	2.61 ± 1.46	0.401	3.85 ± 0.95	1.37 ± 1.71	0.001 **	CTL 3.00 ± 0.25 EXP 3.85 ± 0.18	0.011 *	CTL 2.61 ± 0.34 EXP 1.37 ± 0.33	0.013 *
**Physical reactions**										
Sleep disorder (insomnia or nightmares)	2.06 ± 0.80	1.61 ± 0.98	0.119	2.81 ± 1.11	0.85 ± 1.06	0.001 **	CTL 2.06 ± 0.18 EXP 2.81 ± 0.21	0.011 *	CTL 1.61 ± 0.23 EXP 0.85 ± 0.20	0.018 *
Chronic fatigue (permanent tiredness)	2.00 ± 0.97	1.61 ± 1.09	0.202	2.89 ± 1.15	1.19 ± 1.39	0.001 **	CTL 2.00 ± 0.22 EXP 2.89 ± 0.22	0.008 *	CTL 1.61 ± 0.25 EXP 1.19 ± 0.26	0.257
Headaches or migraines	2.11 ± 1.08	1.94 ± 1.16	0.681	3.00 ± 1.24	1.41 ± 1.55	0.002 **	CTL 2.11 ± 0.25 EXP 3.00 ± 0.23	0.015 *	CTL 1.94 ± 0.27 EXP 1.41 ± 0.29	0.192
Digestive problems, abdominal pain, and diarrhea	1.78 ± 1.06	1.44 ± 0.92	0.302	3.19 ± 1.21	0.67 ± 0.73	0.001 **	CTL 1.78 ± 0.25 EXP 3.19 ± 0.23	0.001 *	CTL 1.44 ± 0.21 EXP 0.67 ± 0.14	0.005 **
Scratching, nail biting, rubbing, etc.	2.39 ± 1.24	1.28 ± 0.89	0.014 **	3.52 ± 1.37	0.67 ± 0.73	0.001 **	CTL 2.39 ± 0.29 EXP 3.52 ± 0.26	0.007 *	CTL 1.28 ± 0.21 EXP 0.67 ± 0.14	0.022 *
Drowsiness or increased need for sleep	2.67 ± 0.91	2.11 ± 1.18	0.096	3.33 ± 1.00	1.22 ± 1.40	0.001 **	CTL 2.67 ± 0.21 EXP 3.33 ± 0.19	0.026	CTL 2.11 ± 0.27 EXP 1.22 ± 0.26	0.027 *
**Psychological reactions**										
Restlessness, inability to relax and be calm	2.56 ± 1.20	1.78 ± 1.17	0.059	3.00 ± 0.88	0.81 ± 0.92	0.001 **	CTL 2.56 ±.28 EXP 3.00 ± 0.16	0.187	CTL 1.78 ±.27 EXP 0.81 ± 0.17	0.013 *
Feelings of depression and sadness	2.00 ± 0.97	1.61 ± 1.09	0.218	2.52 ± 0.75	0.85 ± 1.06	0.001 **	CTL 2.00 ± 0.22 EXP 2.52 ± 0.14	0.065	CTL 1.61 ± 0.25 EXP 0.85 ± 0.20	0.006 **
Anxiety, anguish, despair	2.11 ± 0.90	1.33 ± 0.69	0.001 **	3.26 ± 0.94	0.96 ± 1.09	0.001 **	CTL 2.11 ± 0.21 EXP 3.26 ± 0.18	0.001	CTL 1.33 ± 0.16 EXP 0.96 ± 0.21	0.027 *
Lack of concentration	2.67 ± 1.08	2.50 ± 1.47	0.712	3.33 ± 0.88	1.33 ± 1.54	0.001 **	CTL 2.67 ± 0.25 EXP 3.33 ± 0.16	0.037	CTL 2.50 ± 0.34 EXP 1.33 ± 0.29	0.169
Feelings of aggression or increased irritability	2.11 ± 1.23	2.17 ± 1.20	0.871	2.56 ± 1.01	1.30 ± 1.46	0.002 **	CTL 2.11 ± 0.29 EXP 2.56 ± 0.19	0.213	CTL 2.17 ± 0.28 EXP 1.30 ± 0.28	0.035 *
**Behavioral reactions**										
Conflicts or tendencies to argue or discuss	1.83 ± 0.79	1.94 ± 1.16	0.717	2.52 ± 0.98	1.26 ± 1.43	0.001 **	CTL 1.83 ± 0.18 EXP 2.52 ± 0.18	0.013 *	CTL 1.94 ± 0.27 EXP 1.26 ± 0.27	0.085
Isolation from others	1.78 ± 1.17	2.17 ± 1.47	0.310	2.26 ± 1.02	1.33 ± 1.52	0.019 *	CTL 1.78 ± 0.27 EXP 2.26 ± 0.19	0.164	CTL 2.17 ± 0.34 EXP 1.33 ± 0.29	0.073
Reluctance to complete schoolwork	2.11 ± 0.90	2.50 ± 1.38	0.233	2.44 ± 0.75	1.41 ± 1.50	0.006 **	CTL 2.11 ± 0.21 EXP 2.44 ± 0.14	0.203	CTL 2.50 ± 0.32 EXP 1.41 ± 0.28	0.016 *
Increase or decrease in food consumption	2.67 ± 1.19	1.94 ± 1.35	0.126	3.30 ± 0.91	1.30 ± 1.35	0.001 **	CTL 2.67 ± 0.28 EXP 3.30 ± 0.17	0.066	CTL 1.94 ± 0.31 EXP 1.30 ± 0.26	0.123
**Coping reactions**										
Assertiveness (defending preferences, ideas or feelings without harming others)	3.61 ± 0.70	3.17 ± 1.38	0.238	3.85 ± 0.72	1.85 ± 1.99	0.001 **	CTL 3.61 ± 0.16 EXP 3.85 ± 0.13	0.27	CTL 3.17 ± 0.32 EXP 1.85 ± 0.38	0.012 *
Planning and carrying out tasks	3.28 ± 0.75	2.72 ± 1.36	0.135	3.37 ± 0.79	1.59 ± 1.65	0.001 **	CTL 3.28 ± 0.17 EXP 3.37 ± 0.15	0.694	CTL 2.72 ± 0.32 EXP 1.59 ± 0.31	0.016 *
Self-flattery	3.28 ± 1.07	2.50 ± 1.47	0.110	2.89 ± 1.01	1.11 ± 1.25	0.001 **	CTL 3.28 ± 0.25 EXP 2.89 ± 0.19	0.232	CTL 2.50 ± 0.34 EXP 1.11 ± 0.24	0.002 **
Religiousness (prayers or attendance at mass)	2.22 ± 1.06	2.00 ± 1.33	0.631	2.56 ± 1.19	1.33 ± 1.47	0.422	CTL 2.22 ± 0.25 EXP 2.56 ± 0.22	0.331	CTL 2.00 ± 0.31 EXP 1.33 ± 0.28	0.122
Search for information about the situation	2.61 ± 1.09	2.56 ± 1.38	0.889	3.56 ± 0.97	1.33 ± 1.57	0.001 **	CTL 2.61 ± 0.25 EXP 3.56 ± 0.18	0.006 *	CTL 2.56 ± 0.32 EXP 1.33 ± 0.30	0.009 *
Expression and confidence (talking about the situation of concern)	2.61 ± 1.04	2.67 ± 1.19	0.893	3.30 ± 1.20	1.37 ± 1.50	0.001 **	CTL 2.61 ± 0.24 EXP 3.30 ± 0.23	0.048 *	CTL 2.67 ± 0.28 EXP 1.37 ± 0.28	0.002 *

^(1)^ A Likert-type scale from 1 to 5 was used, where 1 indicated mild to no symptoms and 5 indicated severe symptoms. ^(2)^ The Student’s *t*-test for paired samples was used. ^(3)^ The Student’s *t*-test for independent samples was used. * *p* < 0.05, ** *p* < 0.01.

**Table 3 nutrients-13-01551-t003:** Correlation between stress levels and observed changes in studied microbial phyla, before and after the intervention with BFLAB or placebo beverage.

Treatment	Firmicutes		Bacteroidetes		Gammaproteobacteria	
	Pearson r	*p*	Pearson r	*p*	Pearson r	*p*
CTL before	−0.0899	0.7228	−0.2677	0.2829	−0.0246	0.9228
CTL after	−0.2341	0.1749	−0.3164	0.1004	0.0530	0.8346
CTL before and after (to observe the change over time)	−0.1504	0.1907	−0.2925	0.0834	0.0127	0.9410
EXP before	−0.4268	0.0132 *	−0.2700	0.0866	−0.0261	0.8968
EXP after	−0.2075	0.1496	−0.1520	0.2245	−0.0156	0.9382
EXP before and after (to observe the change over time)	−0.4644	0.0002 *	−0.3894	0.0018 *	−0.0278	0.4231

* *p* < 0.05 indicated significant correlation, using the Pearson test, between stress levels and phyla abundance in feces, as measured with *16S rRNA* gene copy number.

## Data Availability

Data reported in this study can be obtained by contacting Dr. Beatriz Pérez-Armendariz (beatriz.perez@upaep.mx).

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
