# Peer review of "Effect of the Intake of a Traditional Mexican Beverage Fermented with Lactic Acid Bacteria on Academic Stress in Medical Students"

_nutrients, 2021, doi:10.3390/nu13051551_

Round 1

Reviewer 1 Report

The manuscript is very interesting because Authors described effects of ingesting a beverage fermented with Lactic Acid Bacteria (LAB) on stress. Consumption of fermented  beverage significantly increased the phyla Firmicutes and Bacteroidetes comparing to control group. Moreover, the Authors have shown that consumption of fermented food may have a beneficial effect on stress reduction and the correction of he dysbiosis in the gut microbiota. Because the taxonomy is changed and Scientists re-classify the Lactobacillus genus, I suggest to mention about it and also the Authors may include in parentheses new names of L. plantarum, paracasei and brevis.

Author Response

Response to the comments of Reviewer 1

Point 1: Because the taxonomy is changed and Scientists re-classify the Lactobacillus genus, I suggest to mention about it and also the Authors may include in parentheses new names of L. plantarum, paracasei and brevis.

Response 1: The following paragraph was added to the end of the introduction section: “It is important to note that Lactobacillus plantarum, Lactobacillus paracasei and Lactobacillus brevis were recently reclassified as Lactiplantibacillus plantarum, Lacticaseibacillus paracasei, Levilactobacillus brevis respectively [54].”

Reviewer 2 Report

In this manuscript, the authors evaluated the academic stress in medical students after ingestion of a traditional Mexican beverage fermented with Lactobacillus plantarum, Lactobacillus paracasei, and Lactobacillus brevis. Besides, the microbial phyla in feces were quantified by qPCR. This article presents relevant information that could be of interest for readers in the field of microbiology, physiology, nutrition, and medical biology with promising applications that needs to be further investigated. The manuscript is well written and sectioned without remarkable technical or writing issues. 

 Comments:

  • Title: the title is very clumsy and needs to be rewritten
  • Line 25: …108….significantly reduced the …
  • I didn’t see any percentages in the introduction so what was the positive effect of using prebiotics or probiotics on the patient conditions in terms of percentages, sample size, gender, age, significance, etc. I think a comparative table can be included.
  • A figure about the human gut microbiome with few recent information and communication pathway can be included in the introduction section
  • Line 178-180: rewrite using correct English
  • Line 360-363: I see no explanation or hypothesis concerning the observed contradictions
  • The discussion section can be further extended and improved: no statistical comparison in terms of numbers/percentages was indicated when comparing your results. Same when referring to other studies I didn’t find any numbers or statistical comparison.

Author Response

Response to the comments of Reviewer 2

Point 1: Title: the title is very clumsy and needs to be rewritten.

Response 1: The title was revised and the following title is proposed:  

“Effect of the intake of a traditional Mexican beverage fermented with lactic acid bacteria on academic stress in medical students.”

Point 2: Line 25: …108….significantly reduced the …

Response 2: The suggested modifications were implemented in the text. The number 8 was put in superindex form, and the word “significant” was replaced by the word “significantly”. 

Point 3: I didn’t see any percentages in the introduction so what was the positive effect of using prebiotics or probiotics on the patient conditions in terms of percentages, sample size, gender, age, significance, etc. I think a comparative table can be included.

Response 3: Published research articles in this area use various measurement instruments (surveys, biomarkers, etc) and the stress reduction is validated using statistical tests (p<0.05). Thus we followed the reviewer´s advice and added table 1 with comparative data from stress related studies were p values are mentioned.

Point 4: A figure about the human gut microbiome with few recent information and communication pathway can be included in the introduction section.

Response 4: Figure 1 was included in the introduction section.

Point 5: Line 178-180: rewrite using correct English.

Response 5:  The paragraph between lines 178-180: “The instrument used was the SISCO Inventory of Academic Stress. The questionnaire is a proposed and validated instrument by Barraza [49] for the study of the chronical academic stress, measures both behaviour and physical affectation and was applied in other Latin America countries [50,51]”

… was replaced with the following paragraph: “The instrument used in this study was the SISCO Inventory of Academic Stress. This survey is an instrument proposed and validated by Barraza [57] for the study of chronic academic stress. It measures the adverse effect of stress on the behavior and health of students and was previously applied in other Latin American countries [58, 59]”

Point 6: Line 360-363: I see no explanation or hypothesis concerning the observed contradictions.

Response 6: The paragraph around lines 360-363: “The increased abundance of Firmicutes in the gut microbiota of the EXP group after invention with FBLAB may be a consequence of the daily consumption of a beverage fermented with 3 species of the genus Lactobacillus belonging to the phylum Firmicutes (Figure 3). Previous studies have related the consumption of strains of the genus Lactobacillus with increased abundance of Firmicutes in gut microbiota [75,76]. The increase observed in the genus Bacteroidetes after consumption of the FBLAB is in contrast with previous studies which showed that consumption of the strain L. Casei Shirota had no significant effect on this phylum in the gut microbiota of people with obesity [77].”

… was complemented with the following paragraph: “these differences may be due to the fact that in this study subjects suffered from stress and not obesity. The balance of the microbial phyla is multifactorial: gender, stress, obesity, doses, intervention time are some of the factors that modulate the gut micro-biota. In addition, several mechanisms are involved in the balance of the gut microbiota such as the increase of immune cells, the permeability of the intestinal barrier and the presence of antimicrobial peptides [76, 77]. The observed increase in the abundance of both Firmicutes and Bacteroidetes (Figure 4) allows maintaining a balance between these two phyla, a balance usually associated with a healthy gut microbiota [78] and possibly with the observed decrease in stress (Figure 2).”

Point 7: The discussion section can be further extended and improved: no statistical comparison in terms of numbers/percentages was indicated when comparing your results. Same when referring to other studies I didn’t find any numbers or statistical comparison.

Response 7: In the results section: Figure 2 was modified in order to include the percentage of participants that reported each stress level in the control and experimental groups (CTL y EXP). An adequate statistical test (considering that data is reported on a licker scale) was indeed performed in the form of a student T-test. Moreover, the following paragraph was added to the results section: “The results show that in the EXP group 0 % of participants reported high stress levels (N4 and N5) after the intervention, compared to 22.2 % of participants in the CTL group. While 77.8 % of participants in the EXP group reported low stress levels (N1 and N2) after the intervention, compared to 22.2 % of participants in the CTL group.”

 As for the results related to the abundance of the studied phyla shown in figure 3 and 4, the data shown in Figure 4 are the same as data shown in figure 3 but represented in the form of percentage change of abundance of these phyla.

The corresponding paragraph in the results section was improved by including the change percentage values to make this point clearer as follows: “The results are shown in Figure 4. A significant increase in the abundance of Bacteroidetes (83%) was observed after consumption of the FBLAB in the EXP group, but no significant change in the abundance of this phylum (3%) was observed after consumption of the placebo. The consumption of both beverages (FBLAB and placebo) resulted in an increase in the abundance of the phylum Firmicutes for both groups; however, consumption of the FBLAB generated a significantly higher increase (95%) in the abundance of Firmicutes than consumption of the placebo (19%). No significant change was observed in the abundance of the phylum Gammaproteobacteria in gut microbiota after consumption of either beverage (6% for the EXP group and 4% for the CTL group).”

In the discussion section: The results obtained in our study using the SISCO instrument could not be compared to the results obtained in previous studies in terms of numbers or percentages to you to the fundamental differences in the used measurement techniques. However, we added p values to the discussion of the results in order to compare the significance of observed changes. The introduced changes are marked in yellow.  
